# A Mixture of *Cervus elaphus sibiricus* and *Glycine max* (L.) Merrill Inhibits Ovariectomy-Induced Bone Loss Via Regulation of Osteogenic Molecules in a Mouse Model

**DOI:** 10.3390/ijms24054876

**Published:** 2023-03-02

**Authors:** Dong-Cheol Baek, Seung-Ju Hwang, Jin-Seok Lee, Jing-Hua Wang, Chang-Gue Son, Eun-Jung Lee

**Affiliations:** 1Institute of Bioscience & Integrative Medicine, Daejeon Korean Hospital of Daejeon University, Daedukdae-ro 176 bun-gil 75, Daejeon 35235, Republic of Korea; 2Department of Korean Rehabilitation Medicine, Daejeon Korean Hospital of Daejeon University, Daedukdae-ro 176 bun-gil 75, Daejeon 35235, Republic of Korea

**Keywords:** osteoporosis, bone loss, BPX, herbal medicine, bone mineral density, osteogenesis

## Abstract

Osteoporosis is a metabolic skeletal disease characterized by lowered bone mineral density and quality, which lead to an increased risk of fracture. The aim of this study was to evaluate the anti-osteoporosis effects of a mixture (called BPX) of *Cervus elaphus sibiricus* and *Glycine max* (L.) Merrill and its underlying mechanisms using an ovariectomized (OVX) mouse model. BALB/c female mice (7 weeks old) were ovariectomized. From 12 weeks of ovariectomy, mice were administered BPX (600 mg/kg) mixed in a chow diet for 20 weeks. Changes in bone mineral density (BMD) and bone volume (BV), histological findings, osteogenic markers in serum, and bone formation-related molecules were analyzed. Ovariectomy notably decreased the BMD and BV scores, while these were significantly attenuated by BPX treatment in the whole body, femur, and tibia. These anti-osteoporosis effects of BPX were supported by the histological findings for bone microstructure from H&E staining, increased activity of alkaline phosphatase (ALP), but a lowered activity of tartrate-resistant acid phosphatase (TRAP) in the femur, along with other parameters in the serum, including TRAP, calcium (Ca), osteocalcin (OC), and ALP. These pharmacological actions of BPX were explained by the regulation of key molecules in the bone morphogenetic protein (BMP) and mitogen-activated protein kinase (MAPK) pathways. The present results provide experimental evidence for the clinical relevance and pharmaceutical potential of BPX as a candidate for anti-osteoporosis treatment, especially under postmenopausal conditions.

## 1. Introduction

Osteoporosis is a musculoskeletal disease characterized by low bone mass, microstructural destruction of bone, and an increased risk of fractures [1]. According to a meta-analysis of 103, 334, 579 people over the course of 15 years, the global prevalence of osteoporosis was 18.3%, with a prevalence of 23.1% in females and 11.7% in males [2]. More than 3 million cases of osteoporosis and fractures occur annually in the United States, which is expected to account for an economic burden of 25.3 billion dollars by 2025 [3].

In general, primary osteoporosis is divided into type 1 postmenopausal osteoporosis and type 2 senile osteoporosis, while secondary osteoporosis results from a clearly defined etiologic clinical disease or medications [4]. In particular, postmenopausal osteoporosis accounts for 80% of all osteoporosis cases in women over the age of 50 [5]. Estrogen is involved in bone metabolism because it leads to dynamic equilibrium that maintains homeostasis by regulating the balance between osteoclasts and osteoblasts [6]. However, as estrogen levels are reduced during menopause, the rate of bone resorption rises due to an increase in osteoclast differentiation rather than osteoblast differentiation [7,8]. As a result, postmenopausal females have increased bone loss, which leads to osteoporosis, and 40% of postmenopausal females are anticipated to suffer from osteoporotic fractures, including those of the spine, hip, and wrist [9,10].

Currently, as bone resorption inhibitors, bisphosphonates are the primary therapeutic agent used to treat osteoporosis [11,12]. However, long-term use of these medicines has been associated with side effects such as atypical fractures, stroke, and coronary artery disease [13,14,15,16]. Another therapeutic approach is to use agents such as teriparatide, abaloparatide, and romosozumab, which promote bone formation. One clinical trial showed less severe adverse effects in bone formation-promoting agents compared with anti-bone resorption therapy with bone-forming treatment in patients with severe osteoporosis [17]. Recently, as an emerging target for osteoporosis, researchers have focused on the bone morphogenetic protein (BMP) pathway to regulate osteogenic differentiation [18].

Owing to the long clinical use especially in Asian countries and a low risk of side effects, natural resources have recently caught the attention of osteoporosis researchers [19]. According to the latest research, natural resource-based medical approaches for the treatment of osteoporosis have the potential to reduce extremely unbalanced bone turnover, resulting in increased bone mineral density as well as minimal bone microstructural degradation by promoting osteogenesis [20]. Based on long-term clinical experience, we used a standardized water extract syrup (called BPX) of the mixture of *Cervus elaphus sibiricus* and *Glycine max* (L.) Merrill in Daejeon University Hospital, Republic of Korea, since 2020. In a previous study, *Cervus elaphus sibiricus* improved estradiol concentration and femoral bone mineral density (BMD) in the OVX-induced rat model [21,22]. Additionally, treatment with *Glycine max* (L.) Merrill has been previously shown to cause a reduction in the bone resorption marker and an increase in the bone formation marker, as well as an improvement in bone mineral density (BMD) at the proximal femur in 72 postmenopausal females aged 45 to 65 [23].

Meanwhile, no study of the anti-osteoporotic effect of a combination of both resources has been conducted. We herein aimed to investigate the anti-osteoporotic effects of BPX and its related mechanisms using an ovariectomized mouse model.

## 2. Results

### 2.1. Fingerprinting of BPX

Four compounds, uracil, daidzin, glycitin, and genistin, were detected at retention times of 3.53, 17.02, 17.74, and 19.72 min, respectively, in the tested samples. Semiquantitative analysis showed 0.01 mg/g uracil, 0.14 mg/g daidzin, 0.03 mg/g glycitin, and 0.22 mg/g genistin in BPX (Figure 1A–C).

### 2.2. BPX Attenuated OVX-Induced Bone Loss

DXA 2D images showed the notable induction of osteoporosis in OVX-induced mice, which was supported by the decreases in both bone mineral density (BMD) and bone volume (BV) scores. These parameters for osteoporosis were significantly attenuated by administration of BPX compared with the OVX group (*p* < 0.05 or 0.01 for BMD and *p* < 0.05 for BV, Figure 2A–D). These anti-osteoporotic effects were similarly observed in all measurements, including the whole body, femur, and tibia (Figure 2B–D).

### 2.3. BPX Attenuated Histological Alterations in the Femur

H&E staining revealed that OVX reduced the trabecular bone area but increased bone marrow adipocyte volume in secondary spongiosa of the femur, whereas these alterations were remarkably attenuated by BPX administration (Figure 3A,D). The effects of BPX were also notably supported by other histological findings, including staining for both ALP and TRAP activity and the number of osteoblasts and osteoclasts in both cortical and trabecular bone. (Figure 3B,C,E–H). In addition, BPX administration notably attenuated OVX-induced downregulation of OPG while inhibiting RANKL expression and the ratio of RANKL/OPG in the femur (*p* < 0.05 or *p* < 0.01, Figure 3I). 

### 2.4. BPX Attenuated Alterations in Bone Formation and Resorption Markers in Serum

OVX notably elevated serum levels of TRAP, Ca, and OC compared with the Sham group, but administration of BPX markedly attenuated these alterations, especially for the levels of TRAP (*p* < 0.05) and Ca (*p* < 0.01) (Figure 4A–C). BPX administration also significantly attenuated the OVX-induced reduction in the serum ALP level (*p* < 0.05, Figure 4D).

### 2.5. BPX Modulated the BMP Pathway and Its Related Molecules

From osteoblast progenitor cells to matrix mineralization, the proteins and gene expression related to osteoblastogenesis and bone formation was determined. OVX markedly suppressed the protein expression of *p*-p38, *p*-smad 1/5/8, smad 4, and Runx2 in the femur, and BPX administration significantly attenuated these alterations in all molecules (*p* < 0.01, Figure 4F,G). In gene expression analyses, BPX administration significantly normalized the OVX-induced downregulation of BMP-2, BSP-1, and OSX in the femur (*p* < 0.05 or *p* < 0.01, Figure 4E).

## 3. Discussion

Osteoporosis is a metabolic skeletal disease that weakens bones to the point where they break easily, which leads to physical limitations and poor quality of life [24]. As bone resorption inhibitors and antitumor agents, bisphosphonates are the most standard medication for osteoporosis [25,26]. However, it has been reported that there are still some negative effects, so this should be taken with caution [27]. To investigate the anti-osteoporotic potential of BPX and its underlying mechanisms, we employed an OVX-induced mouse model. According to guidelines from the Food and Drug Administration (FDA), this model is an appropriate preclinical model for postmenopausal osteoporosis [28]. OVX is well known to induce estrogen deficiency-related bone loss and the clinical symptoms of postmenopausal osteoporosis [29,30]. As expected, OVX resulted in a decrease in both bone mineral density (BMD) and bone volume (BV) (Figure 2 and Appendix A) as well as the presence of postmenopausal symptoms such as an increase in the levels of FSH in serum and a reduction in uterus size in our present data (Appendix A). BMD is the quantitative parameter of minerals in bone tissue and is used for the diagnosis of osteoporosis [31,32]. A lower BMD, which indicates a loss of bone mass and deterioration of bone tissue, leads to higher bone fragility.

In particular, for females, when estrogen levels become low under postmenopausal conditions, the balance and coupled action of bone resorption and formation are broken, resulting in excessive bone turnover and bone loss [33]. In the first 5 years of menopause, an imbalance in bone remodeling causes continuous and rapid bone loss, primarily in the trabecular bone, followed by affecting cortical bones in the later years [34]. In our study, BPX administration significantly attenuated both the BMD and BV scores in the whole body, femur, and tibia (Figure 2 and Appendix A). The low BMD reflects abnormal bone microarchitecture, which is commonly associated with a decrease in cortical or trabecular bone area and fat accumulation in bone marrow [35,36]. BPX administration notably attenuated the reduction in trabecular bone areas and fat accumulation in the bone marrow of the femur (Figure 3A,D). These results support the anti-osteoporotic effects of BPX under OVX conditions.

These anti-osteoporotic actions of BPX were evidenced by other osteogenic parameters, including two bone resorption markers (TRAP and Ca) and two bone formation markers (OC and ALP) (Figure 4A–D). TRAP is secreted by mature osteoclasts during bone resorption, which decomposes bone tissue, resulting in increased Ca release into blood [37]. Especially under low estrogen conditions, osteoclast cells easily become activated, accelerating bone turnover [38]. Thus, serum levels of TRAP and OC have been reported to be significantly higher in osteoporotic females than in non-osteoporotic postmenopausal females [39]. BPX significantly attenuated the elevation in both TRAP and Ca in blood, which indicates anti-osteoclast activity (Figure 3C,G,H and Figure 4A,B). Furthermore, BPX treatment increased the suppression of ALP in serum and osteoblast activity in the femur (Figure 3B,E,F and Figure 4D). As a specific marker of bone formation in both the early and late stages of osteoporosis, the ALP level is increased during the development of the bone matrix [40]. These results support the mRNA expression of OPG, RANKL, and the RANKL/OPG ratio. BPX administration markedly normalized the suppression of OPG and attenuated the elevation in RANKL and the RANKL/OPG ratio in the femur (Figure 3I). OPG and RANKL are important factors in regulating the activity of osteoblasts and osteoclasts, which is a key part of the dynamic equilibrium between bone resorption and bone formation [41]. The higher the RANKL/OPG ratio, the stronger the bone resorption activity, which directly affects the differentiation of osteoclasts and bone metabolism [42]. OC is known to affect the formation of bone by osteoblasts but also interacts with osteoclasts for bone resorption during bone turnover [43]. In the present study, there was a decreasing tendency of serum OC levels (without statistical significance) by BPX treatment (Figure 4C). 

In a previous pilot study, we confirmed that BPX promoted osteoblast differentiation of MC3T3-E1 (pre-osteoblast cell line) cells through activating ALP (Appendix A). BMP-2 plays a key role in the modulation of osteoblastic bone formation through the canonical BMP/Smad pathway and a non-canonical MAPK pathway [44]. Based on the central role of the BMP pathway in bone formation, we confirmed the effects of BPX on the modulation of BMP-related molecules in the femur (Figure 4F,G). The osteogenic-specific transcription factor Runx2 is activated by both a canonical BMP/Smad pathway and a noncanonical MAPK pathway [44]. The BMP/Smad pathway of mesenchymal stem cells (MSCs) was found to be decreased in subjects with postmenopausal osteoporosis [45], and recombinant human BMP-2 (rhBMP-2) showed positive effects in the OVX-induced mouse model [46] and in a human study of patients with open tibial fractures [47]. We found that BPX administration significantly attenuated the alterations in bone formation-related molecules, such as BMP/Smad (smad 1/5/8 and smad 4), MAPKs (p38), Runx2, and OSX, in the femur (Figure 4E).

From the abovementioned results, we could summarize that the actions of BPX may be linked to both inhibition of bone resorption and promotion of bone formation. HPLC fingerprinting confirmed the major compositional compounds semiquantitatively, including uracil from *Cervus elaphus sibiricus and* isoflavone forms (daidzin, glycitin, and genistin) from *Glycine max* (L.) Merrill (Figure 1A–C). *Cervus elaphus sibiricus* increased the BMD of the tibia and trabecular bone area of the femur in an OVX-induced rat model [48,49]. In a previous study, isoflavone from *Glycine max* (L.) Merrill suppressed bone resorption but activated bone formation in postmenopausal females [50], along with acting on bone formation in an OVX-induced mouse model [51]. These results are consistent with our data and support the anti-osteoporosis effects of BPX; however, limitation of our study is that we could not identify the active compounds in BPX or the change of microstructure, such as connectivity, thickness, and number of trabecular bones.

Taken together, we suggest that BPX is a potential candidate for increasing bone formation and suppressing OVX-induced bone loss in postmenopausal osteoporosis. Its underlying mechanisms may involve increased BMP/Smad and MAPK signaling and bone-specific genes. Further research will be necessary to discover the active compounds of BPX and investigate the detailed underlying mechanisms.

## 4. Materials and Methods

### 4.1. Chemicals and Reagents

Neutral formalin (10%), acetic acid, a bicinchoninic acid (BCA) protein assay kit, ethylenediaminetetraacetic acid disodium salt dihydrate (EDTA), N-(1-naphthyl)-ethylenediamine dihydrochloride, sodium chloride, tetraethyl ethylenediamine (TEMED), Trizma base, Triton X, and Tween 20 were purchased from Sigma-Aldrich (St. Louis, MO, USA).

Additional reagents and chemicals were obtained as follows: 10% ammonium persulfate solution, radioimmunoprecipitation assay (RIPA) buffer, and skim milk were obtained from LPS Solution (Daejeon, Republic of Korea); bovine serum albumin (BSA) was obtained from GenDEPOT (Barker, TX, USA); Mayer’s hematoxylin was obtained from Wako Pure Chemical Industries (Osaka, Japan); Proprep^TM^ was obtained from iNtRON Biotechnology (Seongnam, Republic of Korea); 4% paraformaldehyde (PFA), 10X Tris glycine buffer, and 10X Tris glycine-SDS buffer were obtained from XOGENE (Daejeon, Republic of Korea); protease inhibitor, phosphatase inhibitor, and RNA Later were obtained from Thermo Fisher Scientific (Waltham, MA, USA); methylene alcohol was obtained from Daejung Chemicals & Metals Co. (Siheung, Republic of Korea); polyvinylidene fluoride (PVDF) membranes were obtained from Pall Co. (Port Washington, NY, USA); phospho-p38 (*p*-p38) antibodies were obtained from Santa Cruz Biotechnology (Dallas, TX, USA); suppressor of mothers against decapentaplegic 4 (smad 4) antibodies were obtained from Abcam (Cambridge, MA, USA); phospho-suppressor of mothers against decapentaplegic 1/5/8 (*p*-smad1/5/8), suppressor of mothers against decapentaplegic 1/5/8/9 (smad1/5/8/9), and Runt-related transcription factor 2 (Runx2) antibodies were obtained from Cell Signaling (Danvers, MA, USA); and an actin antibody was obtained from Thermo Fisher Scientific (Waltham, MA, USA). 

### 4.2. BPX Preparation and Fingerprinting

*Cervus elaphus sibiricus* and *Glycine max* (L.) Merrill were purchased from Daehan-Bio pharm (Guri-si, Republic of Korea), and BPX was prepared. All herbs were mixed with 1 L of distilled water and then extracted with boiling water for 2 h. After that, the extracted components of the solution were separated using a Whatman No. 2 filter (Maidstone, UK). Then, the supernatants were filtered again using a Whatman No. 2 filter. The filtrates were concentrated by rotavapor and then lyophilized. These herbal medicine extracts were prepared by mixing them in certain proportions. The final yield of BPX was 10.4% (*w*/*w*). The acquired powders were stored at −80 °C for future use.

Fingerprinting analyses of BPX were conducted using high-performance liquid chromatography (HPLC). A total of 100 mg of BPX and each reference compound (80 μg of uracil and 160 μg of daidzin, glycitin, and genistin) were dissolved in 1 mL of 50% methanol, and the solution was filtered (0.45 μm). Each sample solution was analyzed using a SunFire C18 (5 μm, 4.6 × 250 mm, Waters, MA, USA). The column was eluted at a flow rate of 1 mL/min and a wavelength of 240 to 450 nm using mobile phases A (0.05% phosphate in H_2_O) and B (acetonitrile including phosphate). 

### 4.3. Animals and Ovariectomy

A total of twenty-four female BALB/c mice (7 weeks old, 16–18 g) were purchased from Dae Han Bio Link (Eumseong, Republic of Korea). All animals were housed at room temperature (22 ± 2 °C) and 60 ± 5% relative humidity under a 12 h light:12 h dark cycle and had free access to a commercial pellet diet (Dooyeol-Biotech, Seoul, Republic of Korea) and tap water. The present study was approved by the Institutional Animal Care and Use Committee of Daejeon University (Daejeon, Republic of Korea; Approval No. DJUARB2022-012) and performed according to the Guide for the Care and Use of Laboratory Animals published by the National Institutes of Health (NIH, MD). After acclimation for 7 days, the mice were used for experiments. OVX surgery and experimental design were performed as follows:

Seven-week-old mice were intraperitoneally injected with a ketamine and xylazine mix ture (90 mg/kg), and their skins were shaved. The shaved skins were incised longitudinally to remove the bilateral ovaries. The exposed skin and muscles were closed, and the surgical area was disinfected with povidone-iodine.

### 4.4. Drug Treatment

The mice were randomly divided into 3 groups (*n* = 8 for each group): Sham, OVX, and BPX (OVX + BPX) groups. On the day osteoporosis induction was confirmed, the Sham and OVX groups received a Sham diet, and the BPX group received a BPX diet (1 kg of pellets was treated with 600 mg BPX) for 20 weeks (Figure 1D). The mice were sacrificed at 39 weeks old, and the femur, tibia, uterus, and serum were removed. The femur and tibia were stored at −80 °C. Additionally, to measure the length of the uterus, all the uteri were photographed.

### 4.5. Dual-Energy X-ray Absorptiometry (DXA) Analysis

During the experiment, the bone mineral density (BMD) and bone volume (BV) of the whole body, femur, and tibia were measured repeatedly (0, 8, 12, 16, 20, 24, 28, and 32 weeks) by dual-energy X-ray absorptiometry (DXA) with an InAlyzer (Medikors Co., Seongnam, Republic of Korea). The femur (include in the femoral head form proximal to distal) and tibia (form proximal to distal) were assessed by drawing region of interest (ROI) boxes surrounding the relevant locations using image analysis software. The BMD (bone mineral contents; g/bone area; cm^2^) and BV (cm^3^) were measured for 1 min, and the radiation exposure was 24 s. The measurement was performed five times, and the mean value was determined.

### 4.6. Histological Analysis

Excised femurs and tibias were fixed in 10% neutral formalin for 24 h at room temperature and demineralized with 20% EDTA for four weeks at 4 °C. Then, the femurs were dehydrated using ethanol and xylene and embedded in paraffin. The paraffin blocks were sectioned into 8 μm sections for sagittal slides using a microtome (Leica RM2235, Nussloch, Germany). Mayer’s hematoxylin and eosin (H&E)-stained femur sections were mounted using Canada balsam. In addition, alkaline phosphatase (ALP) and tartrate-resistant acid phosphatase (TRAP) staining procedures were carried out in accordance with the manufacturer’s instructions (MK301, Takara, Japan). The stained sections were observed using an Axio-phot microscope (Carl Zeiss, Jena, Germany). To measure the adipocyte volume/tissue volume (%), adipocyte size (μm^2^) in secondary spongiosa of the femur, and the stain for ALP and TRAP in both cortical and trabecular bone of the femur, all the femurs were photographed, and Ob.S/BS (%) (osteoblast surfaces/bone surface), osteoblast numbers per mm^2^ bone surface, Oc.S/BS (%) (osteoclast surfaces/bone surface), and osteoclast numbers per mm^2^ bone surface were calculated by Image J (NIH).

### 4.7. Enzyme-Linked Immunosorbent Assay (ELISA) 

After sacrificing, serum was immediately collected from the blood by centrifugation at 3000 rpm for 15 min at 4 °C and stored at −80 °C until use. Serum levels of bone turnover markers, that is, osteocalcin (OC) (E-EL-M0864, Elabscience, Houston, TX, USA), alkaline phosphatase (ALP), and tartrate-resistant acid phosphatase (TRAP) (MK301, Takara, Japan), and the biochemical marker calcium (E-BC-K103-M, Elabscience, Houston, TX, USA) were measured using an enzyme-linked immunosorbent assay (ELISA) kit. The procedures were conducted according to the manufacturer’s instructions.

### 4.8. Western Blot Analysis

The left femurs were pulverized into powder using liquid nitrogen with a mortar and pestle, and then RIPA buffer was added. Prepared proteins were separated by 10% polyacrylamide gel electrophoresis and transferred to polyvinylidene fluoride (PVDF) membranes using a Mini-PROTEAN Tetra Cell System (Bio-Rad, Hercules, CA, USA). After blocking in 5% skim milk at room temperature for 1 h, the membranes were incubated with primary antibodies against *p*-p38 (1:1000, sc-166182), p38 (1:1000, ab170099), *p*-smad1/5/9 (1:1000, #13820), smad1/5/8/9 (1:1000, ab13723), smad4 (1:1000, ab40759), Runx2 (1:1000, #12556), and actin (1:1000, MA5-116869) at 4 °C overnight in a shaking plate. After washing with 0.1% TBS-T, the membranes were incubated with HRP-conjugated anti-mouse (1:5000, to detect *p*-p38, p38, and actin) or anti-rabbit (1:5000, to detect *p*-smad1/5/8, smad1/5/8/9, smad4, and Runx2) antibodies for 45 min. The membrane was then developed using an enhanced chemiluminescence (ECL) advanced kit (Thermo Fisher Scientific, Cleveland, OH, USA), and imaging was performed using a FUSION Solo System (Vilber Lourmat, France). Protein expression was semiquantified using Image J (National Institutes of Health, Bethesda, MD, USA).

### 4.9. Quantitative Real-Time PCR Analysis

The gene expression of bone formation markers was determined in femur tissue using real-time PCR. Excised right femurs were pulverized into powder using liquid nitrogen with a mortar and pestle, and then QIAzol reagent (QIAGEN, Hilden, Germany) was added. Total RNA was extracted using QIAzol reagent (QIAGEN, Hilden, Germany), and cDNA was synthesized using a High-Capacity cDNA Reverse Transcription Kit (4368814, Thermo Fisher Scientific, Cleveland, OH, USA). Quantitative real-time PCR was performed using SYBR Green PCR Master Mix (Applied Biosystems, Foster City, CA, USA). Gene expression was analyzed using an IQ5 PCR Thermal Cycler (Bio-Rad Laboratories, Hercules, CA, USA).

The primers were as follows: bone sialoprotein (BSP) (forward: 5′-AAG CAG CAC CGT TGA GTA TGG-3′), (reverse: 5′-CCT TGT AGT AGC TGT ATT CGT CCT C-3′); osterix (OSX) (forward: 5′-AGC GAC CAC TTG AGC AAA CAT-3′), (reverse: 5′-GCG GCT GAT TGG CTT CT-3′); bone morphogenic protein-2 (BMP-2) (forward: 5′-AGC TGC AAG AGA CAC CCT TT-3′), (reverse: 5′-CAT GCC TTA GGG ATT TTG GA-3′); GAPDH (forward: 5′-CAT GGC CTT CCG TGT TCC T′), (reverse: 5′-CCT GCT TCA CCA CCT TCT TGA-3′); receptor activator of nuclear factors κB ligand (RANKL) (forward: 5′-CGA CTC TGG AGA GTG AAG ACA C′), (reverse: 5′-ACC ATG AGC CTT CCA TCA TAG C-3′) and osteoprotegerin (OPG) (forward: 5′-TGT CCA GAT GGG TTC TTC TCA′) (reverse: 5′-CGT TGT CAT GTG TTG CAT TTC C-3′); rotor gene Q software, version 2.3.1.49, from QIAGEN (Hilden, Germany) was used to calculate the relative gene expression. Glyceraldehyde 3-phosphate dehydrogenase (GAPDH) was used as a housekeeping gene.

### 4.10. Cell Culture and Cytotoxicity

Pre-osteoblast (MC3T3-E1) was cultured in Alpha-MEM supplemented with 10% FBS and 1% antibiotic/antimycotic solution. Murine macrophages (Raw 264.7) were cultured in DMEM supplemented with 10% FBS and 1% antibiotic/antimycotic solution. MC3T3-E1 and Raw 264.7 cells were incubated at 37 °C under 5% CO_2_, and the cells (1 × 10^5^ cells/well) were seeded into 96-well microplates and then incubated for 12 h. Then, the cells were pretreated with BPX (25, 50, and 100 μg/mL) for 24 h. To evaluate the cytotoxicity of BPX, we determined the cytotoxicity with a WST-8 assay (EZ-Cytox, DoGen, Republic of Korea). The absorbance at 450 nm was measured using a UV spectrophotometer (Molecular Devices, Sunnyvale, CA, USA) (Appendix A).

### 4.11. ALP Staining and Activity Assay in MC3T3-E1 Cell

The MG-63 cells (4 × 10^4^ cells/well) were seeded into a 6-well plate and then incubated for 12 h. Then the cells were treated with different doses of BPX (25, 50, and 100 µg/mL) or L-ascorbic acid (50 μg/mL) and β-glycerophosphate (10 mM) every 2 days for 7 days. At day 7 after induction, ALP staining procedures were carried out in accordance with the manufacturer’s instructions (MK301, Takara, Tokyo, Japan). ALP-positive cells were stained blue/purple and observed using an Axio-phot microscope (Carl Zeiss, Germany). The ALP activity was measured using 1-StepTM p-nitrophenyl phosphate substrate solution (Thermo Fisher Scientific, Cleveland, OH, USA). The absorbance was read at 405 nm using a UV spectrophotometer (Molecular Devices, CA, USA). (Appendix A).

### 4.12. Statistical Analysis

The results were expressed as the mean ± standard deviation (SD) or fold changes in the means. Statistical significance was determined by using one-way analysis of variance (ANOVA) followed by Dunnett’s test. In all analyses performed using GraphPad Prism 7 (GraphPad Software, San Diego, CA, USA), *p* < 0.05 was considered to indicate statistical significance.

## Figures and Tables

**Figure 1 ijms-24-04876-f001:**
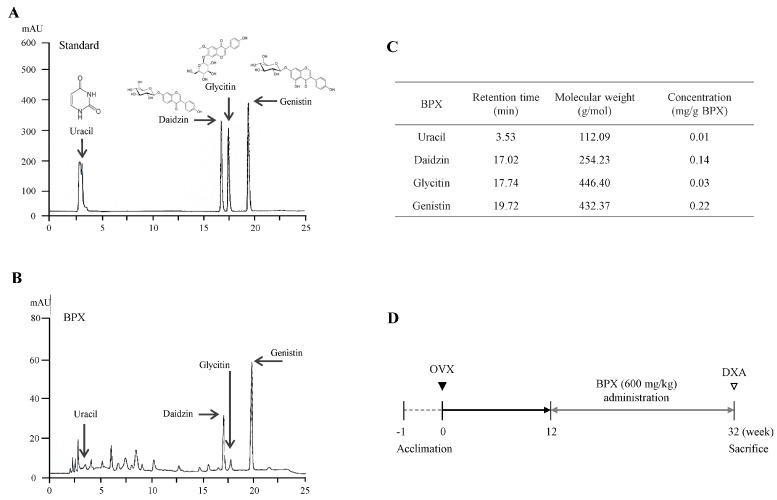
Fingerprinting analysis of BPX and design of the animal study. Chemical constitutions and quantitative analysis of BPX using high-performance liquid chromatography (HPLC). Six reference standards (**A**) and BPX (**B**) were subjected to UHPLC analysis. A quantitative analysis of BPX was conducted (**C**). Schematic of the experimental design (**D**).

**Figure 2 ijms-24-04876-f002:**
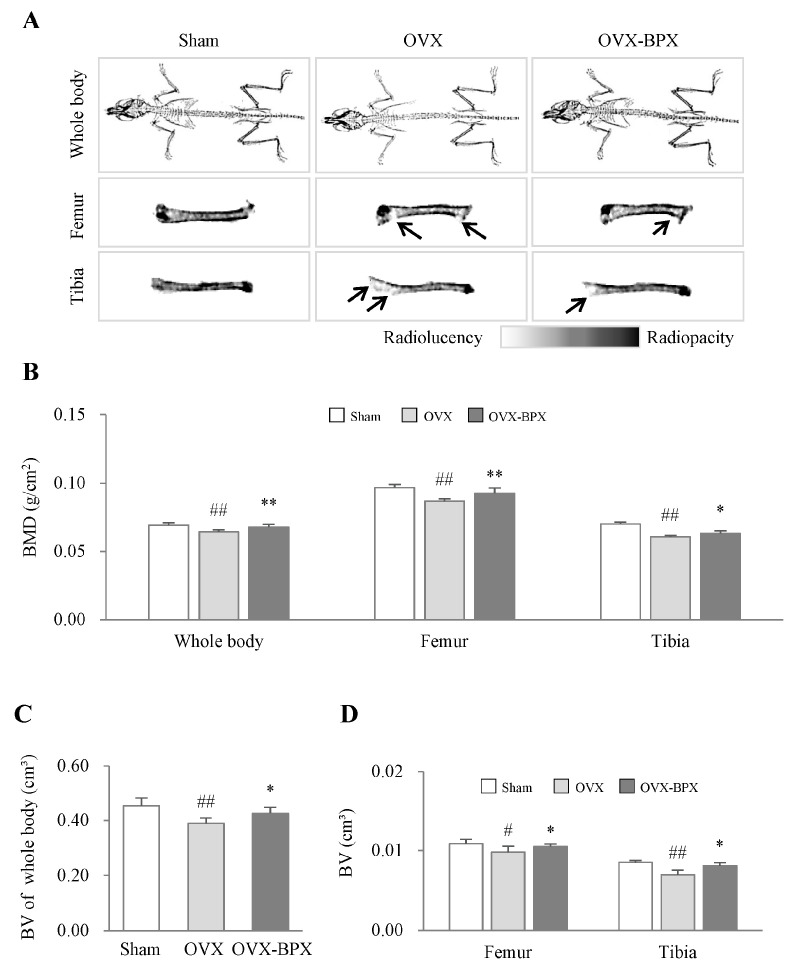
Inhibitory effects of BPX on OVX-induced bone loss. Representative 2D DXA images of the whole body, femur, and tibia (**A**). Scores of BMD in the whole body, femur, and tibia (**B**) and BV in the whole body (**C**), and femur and tibia (**D**) were analyzed by DXA in OVX-induced mice. The mice were divided into groups according to the treatment: the Sham, OVX, and BPX (OVX + BPX) groups. The data are expressed as the mean ± SD. ^#^ *p* < 0.05, ^##^ *p* < 0.01 compared with the Sham group; * *p* < 0.05, ** *p* < 0.01 compared with the OVX group.

**Figure 3 ijms-24-04876-f003:**
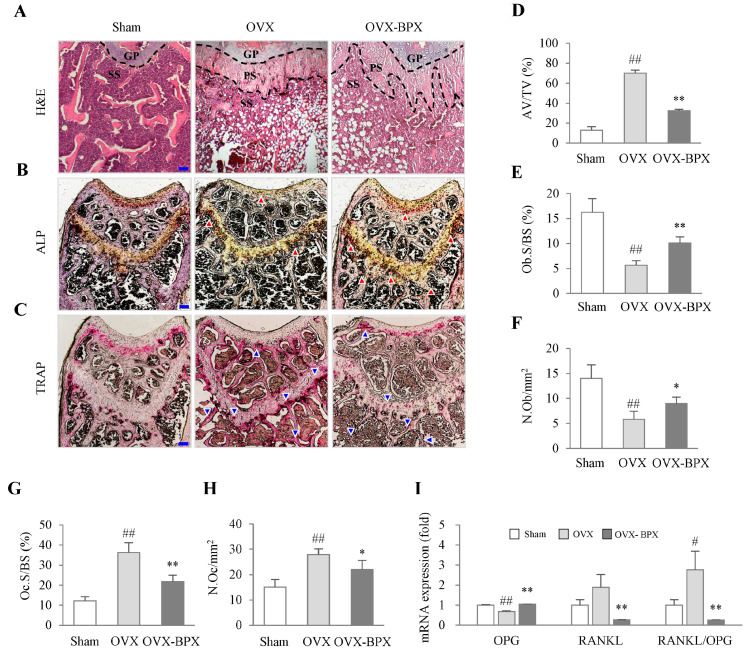
Effects of BPX on histological alterations in the femur. Evaluation of histological alterations using H&E staining (**A**), ALP staining (**B**), and TRAP staining (**C**) was performed, and representative photographs (40× magnification) were evaluated for AV/TV (%) (bone marrow adipocyte volume/tissue volume) at secondary spongiosa (**D**), Ob.S/BS (%) (osteoblast surfaces/bone surface) (**E**), osteoblast numbers per mm bone surface (**F**), Oc.S/BS (%) (osteoclast surfaces/bone surface) (**G**), osteoclast numbers per mm bone surface (**H**), and OPG, RANKL, and RANKL/OPG ratio (**I**) in the femur in OVX-induced mice. The bone area, ALP, and TRAP staining areas were quantified by ImageJ. The mice were divided into groups according to the treatment: the Sham, OVX, and BPX (OVX + BPX) groups. GP = growth plate, PS = primary spongiosa, and SS = secondary spongiosa. The data are expressed as the mean ± SD. ^#^ *p* < 0.05, ^##^ *p* < 0.01 compared with the Sham group; * *p* < 0.05, ** *p* < 0.01 compared with the OVX group.

**Figure 4 ijms-24-04876-f004:**
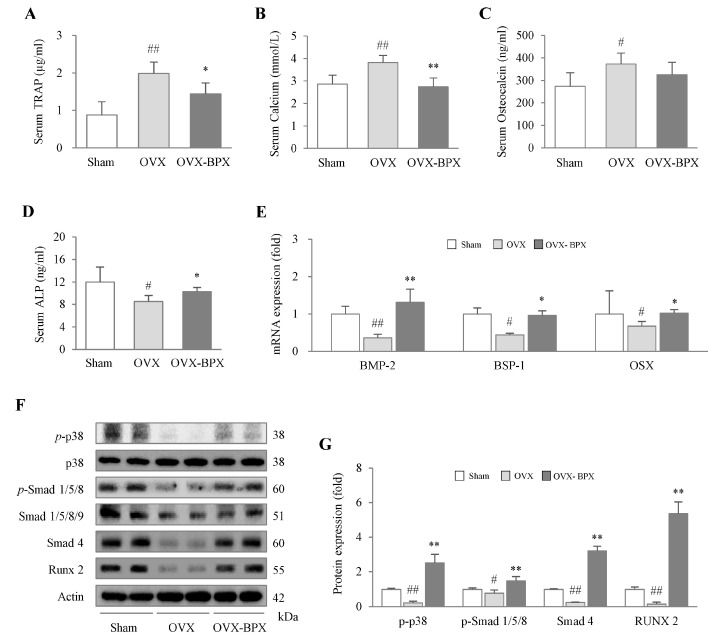
Modulatory effects of BPX on bone formation and resorption markers in serum and modulation of BMP pathways in the femur. The levels of bone resorption markers, such as TRAP (**A**) and Ca (**B**), and bone formation markers, such as osteocalcin (**C**) and ALP (**D**), in serum were analyzed by ELISA. The mRNA expression of BMP-2, BSP-1, and OSX were assessed (**E**), and the protein levels of *p*-p38, *p*-smad 1/5/8, smad 4, and Runx2 were assessed by western blot (**F**,**G**) in the femurs of OVX-induced mice. All band intensities were quantified by Image J. The mice were divided into groups according to the treatment: the Sham, OVX, and BPX (OVX + BPX) groups. The data are expressed as the mean ± SD. ^#^ *p* < 0.05, ^##^ *p* < 0.01 compared with the Sham group; * *p* < 0.05, ** *p* < 0.01 compared with the OVX group.

## Data Availability

The original contributions presented in the study are included in the article/Supplementary Material, further inquiries can be directed to the corresponding authors.

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
