# Peer review of "A Mixture of Cervus elaphus sibiricus and Glycine max (L.) Merrill Inhibits Ovariectomy-Induced Bone Loss Via Regulation of Osteogenic Molecules in a Mouse Model"

_ijms, 2023, doi:10.3390/ijms24054876_

Round 1
Reviewer 1 Report
This is a well written and exciting study that presents data of great interest into those working in the field of osteoporosis and other bone disorders. Authors have nicely presented the evidence of showing the effects of compound BPX in inhibition of OVX-mediated bone loss. As currently available bone targeted therapies are associated with moderate to severe side effects thus any compound which has beneficial effects on bone loss without minimal side effects can be promising in treatment of several bone disorders. Authors have tried to explain the mechanisms of BPX mediated inhibition of bone loss, however there are several aspects need to clarify to strengthen the manuscript.
· Introduction: Lines 49-52, reference 13-16: Bisphosphonates have antitumor effects in breast cancer cells. Authors need to careful by generalizing that bisphosphonates are associated with the side effects such as breast cancer. In the reference that authors have provided is talking about associated of hormonal therapies with breast cancer. Statement related bisphosphonate needs to present with caution as its antitumor effects are well defined.
· Result section, line 78: there are four compounds presented not three. Also, the unit of time is not mentioned. Is it minutes of hours or days?
· Page 4: DXA/Xray bone images are not clear. High resolution images are required to see the differences in the BMD and BV with clear labelling using arrows at the areas of bone loss.
· Page5: Figure 2A, BPX showed reduced bone marrow adiposity but adipocyte size looked increased. Adipocyte size is an important parameter to consider for determining the adiposity. Authors are requested to determine the size of the adipocytes. It is unclear if the bone marrow adiposity is normalized with marrow area/volume
· Page 5: Figure 3: please add arrows to indicate the changes. Overall resolution of the figures is poor. High resolution images are required.
· Page 5: Figure 3B: ALP images are not clear, high resolution images are required. As authors have claimed that BPX increasing bone formation thus presenting osteoblast numbers or activity is critical to support this claim
· Page 5: Figure 3C, TRAP images show non-specific staining at several places. Counting of TARP positive osteoclast numbers at bone surface is gold standard and best way of representing the osteoclast activity. Quantification of % TRAP staining area does not look accurate due to nonspecific staining.
· Figure 3: Presentation of % positive staining area is not sufficient, counting of adipocytes, osteoblasts and osteoclast activity is necessary
· Page 5: Line 124: Please provide rationale for selecting specific bone markers in western blot and gene analysis data.
· Supplemental figure 1C: BPX effects are not remarkable and possible reason could be the time of sample collection. It is unclear if FSH was measured from the serum collected at the same time.
· Supplemental figure 1D, does not show increased body weight in OVX mice. However, on page 6, lie 149 indicates weight gain with OVX. Authors are required to clarify this
· It is unclear if normal mice have undergone Sham-OVX surgery to match the OVX
· Since Normal group is not treated with BPX, thus it is unclear if BPX has any effects on normal bone remodeling or it works in disease state. For preliminary study like this it is necessary to show the effects of drug on normal bone.
· Authors have suddenly started talking about BMP pathway in discussion. There are no mechanisms to show that the increased bone formation is due to increase osteoblast activity or decreased osteoclast activity. Without identifying the basic mechanisms of inhibition of bone loss in this model authors have started focusing on osteoblasts mediated downstream signaling. Clear rationale for selecting osteoblast specific pathway is missing. Solid evidence is requiring for selecting molecules of osteoblast pathway.
· BMD data is the central point of the entire manuscript. It is not clear from the method, result or discussion that what BV is and how it has been calculated. Authors have mentioned about ROI selection, but which region of tibia or femur was selected to define ROI. More clarity about BMD analysis is needed.
· Changes in bone using microCT can provide better understanding of the bone microarchitecture and thus if bones are still available then microCT data can be included.
· It is unclear that how the treatment time period was determined.
· Nomenclature of groups is confusing. Normal, control looks the same, however control is OVX. It would be ideal to label as Sham-OVX, OVX-Vehicle, OVX+BPX or something in that line.
· In method: Details are needed for the DXA analysis parameters and selection of region for ROI drawing.
Author Response
- Introduction: Lines 49-52, reference 13-16: Bisphosphonates have antitumor effects in breast cancer cells. Authors need to careful by generalizing that bisphosphonates are associated with the side effects such as breast cancer. In the reference that authors have provided is talking about associated of hormonal therapies with breast cancer. Statement related bisphosphonate needs to present with caution as its antitumor effects are well defined.
â–º We sincerely appreciate for reviewer’s helpful comments. As reviewer mentioned, we fully agree with reviwer’ professional advice that we need to be careful by generalizing that bisphosphonates are associated with side effects. Thus, we deleted the related in ‘Introduction section’ (Line 51-52) and described with references in "Discussion" section (Line 147-150) of revised manuscript.
- Result section, line 78: there are four compounds presented not three. Also, the unit of time is not mentioned. Is it minutes of hours or days?
â–º We really apologize for the our mistake. As reviewer mentioned, we analyzed that BPX have four chemical compositions, including uracil (in Cervus elaphus sibiricus), daidzin, glycitin, and genistin (in Glycine max L. Merrill). We displayed unit of time in and corrected typing error in ‘Result section’ of revised manuscript (Line 78-79).
- Page 4: DXA/Xray bone images are not clear. High resolution images are required to see the differences in the BMD and BV with clear labelling using arrows at the areas of bone loss.
â–º Thank reviewer for the helpful comments. According to reviewer suggestions, we improved the image resolution and inserted arrows to discriminate the differences of mineral bone density in Figure. 2A.
- Page5: Figure 2A, BPX showed reduced bone marrow adiposity but adipocyte size looked increased. Adipocyte size is an important parameter to consider for determining the adiposity. Authors are requested to determine the size of the adipocytes. It is unclear if the bone marrow adiposity is normalized with marrow area/volume
â–º Thank you for the constructive comment. According to reviewer suggestions, we measured the adipocyte size in bone marrow using Image J (NIH). However, BPX did not significantly increase adipocyte size compared with the OVX group. The extra data is added in Supplementary Figure. 5. Besides, we re-analyzed the bone marrow adiposity per tissue volume (%), and these corrections were reflected in revised Figure. 3D and manuscript.
- Page 5: Figure 3: please add arrows to indicate the changes. Overall resolution of the figures is poor. High resolution images are required.
â–º Thank you for pointing it out. We added arrows and improved the image resolution in Figure 3.
- Page 5: Figure 3B: ALP images are not clear, high resolution images are required. As authors have claimed that BPX increasing bone formation thus presenting osteoblast numbers or activity is critical to support this claim
â–º We fully agree with reviwer’ professional opinions. Based on knowledge for meaning ALP on osteoblast activity, we intended to show the osteoblast activity through ALP-stained levels. Despite of our intention, we re-analyzed the activity of ALP-stained osteoblast per bone surface, and these corrections were reflected in revised Fig. 3E and F and manuscript.
- Page 5: Figure 3C, TRAP images show non-specific staining at several places. Counting of TARP positive osteoclast numbers at bone surface is gold standard and best way of representing the osteoclast activity. Quantification of % TRAP staining area does not look accurate due to nonspecific staining.
â–º Thank you for the constructive suggestion. We also felt empathy for the reviewer's suggestion that the best way to represent osteoclast activity was by counting TRAP-positive osteoclast activity. we re-analyzed the activity of TRAP-stained osteoclast per bone surface, and these corrections were reflected in revised Fig. 3G and H and manuscript. In addition, To support the inhibitory effects of BPX on the TRAP and osteoclast activity, we additionally analyzed the mRNA levels of OPG, RANKL, and the RANKL/OPG ratio in the femur The extra data is added in Figure. 3I.
- Figure 3: Presentation of % positive staining area is not sufficient, counting of adipocytes, osteoblasts and osteoclast activity is necessary
â–º Thank you for the constructive advice. We revised it in accordance with reviewer suggestions (Figure. 3).
- Page 5: Line 124: Please provide rationale for selecting specific bone markers in western blot and gene analysis data.
â–º Thank you for the constructive comment. The rationale for selecting specific bone markers is briefly described in the revised manuscript's "Result" section (Line 129-130) and described in detail with references in "Discussion" section (Line 197-200)
- Supplemental figure 1C: BPX effects are not remarkable and possible reason could be the time of sample collection. It is unclear if FSH was measured from the serum collected at the same time.
â–º Thanks a lot for the professional comment. The serum was collected at the same time (32 weeks after OVX at 2:00 P.M.). Rather, since there is no significant effect of BPX on serum FSH levels, it can be speculated that BPX inhibits OVX-induced bone loss through an increase in molecules involved in bone formation rather than hormonal regulation. Further research will be necessary to discover the various concentrations of BPX that have effects.
- Supplemental figure 1D, does not show increased body weight in OVX mice. However, on page 6, lie 149 indicates weight gain with OVX.
â–º Sorry for the mistake. We revised the discussion (Line 155-158).
- Authors are required to clarify this
- It is unclear if normal mice have undergone Sham-OVX surgery to match the OVX
â–º Sorry for making confusion. Group-naming ‘Normal’ mice are ‘sham-operated mice’ (placebo surgery). We have changed group-naming ‘Normal’ into ‘Sham’, ‘Control’ into ‘OVX’ and ‘BPX’ into ‘OVX-BPX’ in entire manuscript and Figures.
- Since Normal group is not treated with BPX, thus it is unclear if BPX has any effects on normal bone remodeling or it works in disease state. For preliminary study like this it is necessary to show the effects of drug on normal bone.
â–º Thank you for pointing it out. In a previous pilot study, we determined that the BPX treatment had no effect on bone mineral density in normal mice and it works in disease state. The extra data is added in Supplementary Figure. 4.
- Authors have suddenly started talking about BMP pathway in discussion. There are no mechanisms to show that the increased bone formation is due to increase osteoblast activity or decreased osteoclast activity. Without identifying the basic mechanisms of inhibition of bone loss in this model authors have started focusing on osteoblasts mediated downstream signaling. Clear rationale for selecting osteoblast specific pathway is missing. Solid evidence is requiring for selecting molecules of osteoblast pathway.
â–º We sincerely appreciate for reviewer’s advice. In a previous pilot study, we confirmed that BPX promoted osteoblast differentiation of MC3T3-E1 cells through activating alkaline phosphatase (ALP). Thus, we selected osteoblast differentiation and bone formation-related molecules through the BMP pathway. The extra data is added in Supplementary Figure. 3C and D.
- BMD data is the central point of the entire manuscript. It is not clear from the method, result or discussion that what BV is and how it has been calculated. Authors have mentioned about ROI selection, but which region of tibia or femur was selected to define ROI. More clarity about BMD analysis is needed
â–º Thank you for pointing it out. Details of BMD calculation methods were added in Method section as follows: The volume of mineralized bone per unit volume of sample is referred to as "bone volume" (BV; cm3). We defined ROI for the femur as femoral head from proximal to distal, and for the tibia as from proximal to distal. Descriptions for the ROI definition were written in ‘Materials and Method section’ with displaying in Supplementary Figure. 2A.
- Changes in bone using micro CT can provide better understanding of the bone microarchitecture and thus if bones are still available then micro CT data can be included.
â–º Thank you for the constructive suggestion. We also believe that Micro-CT data would support the our findings. Unfortunately, we don't have a statistically constant number of bones available for micro-CT. The BMD data obtained from DXA analysis is sufficient to explain osteoporosis and effects of BPX. In further study, we will try to determine the bone microarchitecture to obtain more convincing outcomes.
- It is unclear that how the treatment time period was determined.
â–º We sincerely appreciate the reviewer’s pointing it out. In general, BPX is prescribed to patients with osteoporosis or osteopenia for 6 to 12 months in our Oriental hospital. The duration of treatment was determined to be 20 weeks based on the human lifespan equivalent to that of mice.
- Nomenclature of groups is confusing. Normal, control looks the same, however control is OVX. It would be ideal to label as Sham-OVX, OVX-Vehicle, OVX+BPX or something in that line.
â–º Sorry for making confusion. We have changed group-naming ‘Normal’ into ‘Sham’, ‘Control’ into ‘OVX’ and ‘BPX’ into ‘OVX-BPX’ in entire manuscript and Figures according to the reviewer’s suggestion.
- In method: Details are needed for the DXA analysis parameters and selection of region for ROI drawing.
â–º We sincerely appreciate for reviewer’s advice. The femur (including in the femoral head from proximal to distal) and tibia (from proximal to distal) were assessed by drawing region of interest (ROI) boxes surrounding the relevant locations using image analysis software. The BMD (Bone mineral contents; g / bone area; cm2) and BV (cm3) were measured for 1 minute, and the radiation exposure was 24 seconds. We described the related contents in the methods portion according to the reviewer’s suggestion (Line 293-297) and the extra data is added in Supplementary Figure. 2A.

Reviewer 2 Report
In this manuscript, the authors investigated BPX, a mixture of water-boiled extracts from Cervus elaphus sibiricus and Glycine max (L.) Merrill, to inhibit ovariectomy-induced bone loss. They found that ovx-reduced BMD and BV, -elevated bone resorption markers, and -decreased bone formation markers were significantly altered after BPX treatment. They found the BMP non-canonical-Smad signaling pathway may be involved BPX inhibiting ovx-induced bone loss. This manuscript is easy to follow, and the data is convincing. According to the findings, the authors suggest that BPX can be a potential candidate for increasing bone formation and suppressing OVX-induced bone loss in postmenopausal osteoporosis.
The authors' findings are worth publishing, however, in my opinion, this manuscript does not qualify for publishing in IJMS because this study is not novel, and the molecular mechanism study is only preliminary results. Authors should do more pathways investigation to understand the mechanisms that BPX possibility involved in.
Comments:
1. Were the normal group mice subjected to sham ovariectomy?
2. In this manuscript, the authors showed 2D DXA images of the skeleton in figure 2, it would be better to support the authors' results had micro-CT images for spines or long bones.
3. Did the TGF-beta/Smad pathways are also involved in the mechanism of BPX?
5. Do Rankl and Opg mRNA and proteins change after BPX treatment in sham or ovx mice?
6. Why did not include BPX-treated sham-ovx mice in this study design to confirm whether BPX has side effects, and also to observe whether BPX can increase bone mineral density in normal mice?
7. In line 78: Should be four compounds?
8. Suggest authors change the control group to ovx group, BPX group to ovx-BPX that will be clearer.
Author Response
- Were the normal group mice subjected to sham ovariectomy?
â–º We are really sorry for making confusion. Group-naming ‘Normal’ mice are ‘sham-operated mice’ (placebo surgery). We have changed group-naming ‘Normal’ into ‘Sham’, ‘Control’ into ‘OVX’ and ‘BPX’ into ‘OVX-BPX’ in entire manuscript and Figures.
- In this manuscript, the authors showed 2D DXA images of the skeleton in figure 2, it would be better to support the authors' results had micro-CT images for spines or long bones.
â–º We fully agree with reviwer’ professional comments. We also believe that Micro-CT data would definitely support the our findings. Unfortunately, we don't have a statistically constant number of bones available for micro-CT. In this study, the BMD data obtained from DXA analysis is sufficient to explain osteoporosis and effects of BPX. We will further investigate bone microarchitecture using micro-CT to obtain more convincing outcomes.
- Did the TGF-beta/Smad pathways are also involved in the mechanism of BPX?
â–º We sincerely appreciate for reviewer’s questions. In a previous pilot study, we confirmed that BPX did not affect the TGF-beta/Smad pathways (Smad 2 and 3).
- Do Rankl and Opg mRNA and proteins change after BPX treatment in sham or ovx mice?
â–º In order to address for reviewer’ question, we additionally analzyed the mRNA levels of OPG, RANKL, and the RANKL/OPG ratio in the femur. Results showed that BPX administration notably normalized OVX-induced downregulation of OPG while inhibiting RANKL expression and the ratio of RANKL/OPG. we described the related contents in ‘Result section’ of revised manuscript and the extra data is added in Figure. 3I.
- 5. Why did not include BPX-treated sham-ovx mice in this study design to confirm whether BPX has side effects, and also to observe whether BPX can increase bone mineral density in normal mice?
â–º Thank you for pointing it out.
- In a previous pilot study, we determined that the BPX treatment had no effect on bone mineral density in sham or normal mice and it works in disease state. The extra data is added in Supplementary Figure. 4.
- To check the toxicity of BPX, we additionally performed a cell viability test. Even the high concentrations of BPX (200 μg) also do not show cytotoxicity in MC3T3-E1 (pre-osteoblast) and Raw 264.7 cells. The extra data is added in Supplementary Figure. 3A and B.
- In line 78: Should be four compounds?
â–º We really apologize for the our mistake. As reviewer mentioned, we analyzed that BPX have four chemical compositions, including uracil (in Cervus elaphus sibiricus), daidzin, glycitin, and genistin (in Glycine max L. Merrill). We displayed unit of time in and corrected typing error in ‘Result section’ of revised manuscript.
- Suggest authors change the control group to ovx group, BPX group to ovx-BPX that will be clearer.
â–º Thank you for the constructive suggestion. We have changed group-naming ‘Normal’ into ‘Sham’, ‘Control’ into ‘OVX’ and ‘BPX’ into ‘OVX-BPX’ in entire manuscript and Figures.

Reviewer 3 Report
This is an interesting article where they look at BPX in preventing estrogen deficiency associated bone loss
I think the paper would be greatly improved if they included micro-CT analysis of bone microarchitecture parameters. It is currently Difficult to discern whether there affects of BPX is in the cortical or the trabecular bone
Author Response
- I think the paper would be greatly improved if they included micro-CT analysis of bone microarchitecture parameters. It is currently Difficult to discern whether there affects of BPX is in the cortical or the trabecular bone
â–º We fully agree with reviwer’ professional comments. We also believe that Micro-CT data would definitely support the our findings. Unfortunately, we don't have a statistically constant number of bones available for micro-CT. In this study, the BMD data obtained from DXA analysis is sufficient to explain osteoporosis and effects of BPX. Micro-CT shows the interior structure of mostly trabecular bone, whereas our DXA analyzes the bone mineral density of both cortical and trabecular bone structures.
Based on our knowledge of the impact of osteoporosis on bone mineral density, we intended to show the effects of both the cortical and trabecular bones through DXA-bone mineral density analysis. We will further investigate bone microarchitecture using micro-CT to obtain more convincing outcomes.

Round 2
Reviewer 1 Report
Thank you for adding the requested changes. However, all the concerns are not sufficiently addressed. High resolution images of Xray, histology is not included. For histology images, growth plate is included. In osteoporosis, trabecular bone loss is evident and thus the region of interest should be secondary spongiosa. If authors could replace the histology images by changing region of interest that would better reflect the study outcome in terms of proposed drug effect. In addition to the fact that microCT data is critical for such a study, DXA alone is insufficient. If the author could add that a paragraph describing the limitations of the study would be helpful. Xray images are with very poor resolution and not of publication quality, improvement in it is necessary.
Author Response
Thank you for adding the requested changes. However, all the concerns are not sufficiently addressed. High resolution images of Xray, histology is not included. For histology images, growth plate is included. In osteoporosis, trabecular bone loss is evident and thus the region of interest should be secondary spongiosa. If authors could replace the histology images by changing region of interest that would better reflect the study outcome in terms of proposed drug effect. In addition to the fact that micro-CT data is critical for such a study, DXA alone is insufficient. If the author could add that a paragraph describing the limitations of the study would be helpful. Xray images are with very poor resolution and not of publication quality, improvement in it is necessary.
â–º We sincerely appreciate for reviewer’s helpful comments and really apologize for our mistake.
- As reviewer mentioned, we fully agree with reviewer’ professional advice that we need high-resolution images of Xray and histology. Thus, we improved the image resolution in entire Figures.
- We also felt empathy for the reviewer's suggestion and we also felt empathy for the reviewer's suggestion that trabecular bone loss is evident in the secondary spongiosa of trabecular bone in osteoporosis. However, our histology images (H&E) revealed the growth plate and also the secondary spongiosa in the trabecular bone and we analyzed bone marrow adipocyte volume/tissue volume at secondary spongiosa. Based on our knowledge of the impact of osteoporosis on bones, we intended to show the effects of trabecular bones through histology (H&E). (Further, our findings with ALP and TRAP staining show that BPX affects both cortical and trabecular bone structures.)
As, reviewer mentioned, we revised histology images by indicating the region (GP = growth plate, PS = primary spongiosa and SS = secondary spongiosa.) and related results described in the revised manuscript's "Result" section (Line 103-105), "Materials and Methods" section (Line 316-318) and figure legend.
- As reviewer mentioned, we also believe that micro-CT data would support our results. However, we could not apply the change of bone microstructure, such as connectivity, thickness, and number of trabecular bones. Thus, these limitations of our study are described in the "Discussion Section" of the revised manuscript. (Line 226-228). Further research will definitely be necessary to discover the effects of BPX on the microstructure of the trabecular bone.

Reviewer 2 Report
no more comment.
Author Response
- no more comment.
â–º We’re really grateful for reviewer's help.

Reviewer 3 Report
Looks ready for publication
Author Response
- Looks ready for publication
â–º We really appreciate for reviewer.
